# Evaluation of microbiome and physico-chemical profiles of fresh fruits of *Musa paradisiaca, Citrus sinensis* and *Carica papaya* at different ripening stages: Implication to quality and safety management

**Dawit Raga Kifle**[1,2]*, **Ketema Bedanie Bacha**[1], **Reda Nemo Hora**[3], **Lata Lachisa Likasa**[1]

**1** Department of Biology, College of Natural Sciences, Jimma University, Jimma, Ethiopia, **2** Department of Biology, College of Natural and Computational Sciences, Mizan-Teppi University, Teppi, Ethiopia, **3** Department of Biology, College of Natural and Computational Sciences, Dembi Dollo University, Dembi Dollo, Ethiopia

* dawitk.galo@gmail.com

**Data Availability Statement:** All relevant data are within the paper and its Supporting Information files.

## Abstract

### Introduction

The ripening of fleshy fruits is a developmental process that involves changes in color, texture, aroma, nutrients, and diversity of microbiomes. Some microorganisms, specially, bacteria and molds are responsible for postharvest spoilage of fruits. Thus, this study is aimed at evaluating the alterations in microbiome and physico-chemical properties of selected fruits at different ripening stages.

### Methods

Totally, 108 fresh fruit samples of *Musa paradisiaca, Citrus sinensis* and *Carica papaya* at three ripening stages were collected and processed in this study. The biochemical methods and MALD-TOF MS were used in identification. The physico-chemical properties of all samples were analyzed using standard methods.

### Results

The minimum counts (6.74± 0.48–6.76± 0.42 log CFU/mL) and the maximum count (7.51± 0.43–7.96± 0.34 log CFU/mL) of AMB in all fruit samples was observed at mature green and overripe stages of the fruits, respectively. The ripening stage has significantly affected the microbial counts (P < 0.05) in all fruits, except counts of *Enterobacteriaceae* in banana and orange, and fungal counts in orange. The bacterial community of all fruits was predominated by *B. cereus* (33.7%), *A. faecalis* (17.3%), *P. putida* (15.2%), *M. morganii* (11.1%), *S. sciuri* (6.6%) and *S. epidermidis* (4.9%); while the fungal microbiome was constituted by *Candida* spp. (33.9%) followed by *Saccharomyces* spp. (18.1%) and *Aspergillus* spp. (16.3%). The ripening stages have also significantly affected the physico-chemical property in all samples. Accordingly, the lowest pH (3.53) and highest content of ascorbic acid (69.87 mg/

**Funding:** The work was undertaken as part of megaproject funded by Jimma University, College of Natural Sciences, under the grant agreement (CNS-BIOL-02/2020) to KB. The funders had no role in study design, data collection and analysis, decision to publish, or preparation of the manuscript.

**Competing interests:** The authors have declared that no competing interests exist.

100g) were observed in mature green oranges and overripe papaya, respectively, while the maximum concentration of total sugar (17.87%) and reducing sugar (14.20%) were recorded in overripe bananas.

## Conclusion

The presence of some potential human pathogens and spoilage microorganisms in fruit samples could contribute to post-harvest product losses besides the potential health risk associated with consumption of the tainted fruits. Hence, proper safety management practices and preservation mechanisms should be developed and put in place to ensure consumers safety against pathogens besides minimizing product losses through microbial spoilage.

## Introduction

True fruits are specialized plant organs found solely in angiosperms, and these unique organs are believed to have evolved to improve seed protection and dispersal by creating attractive flesh and aroma for seed-dispersing animals [1]. Fruits are important for human nutrition, providing energy, carbohydrates, essential amino acids, minerals, fibers, vitamins and antioxidants essential for human nutrition and health [2]. The issue of food losses is of high importance in the efforts to combat hunger, raise income and improve food security in the world's poorest countries [3]. The world is producing annually 675 million tons of fruits to meet the nutrition requirement of its population as per the statistics available in 2017 [4]. Yet, huge post-harvest losses reach in the range of 15–20% in the USA and Canada, 25% in Europe, and 30–35% in Asia and an estimated one in eight people suffer from chronic under-nourishment [5]. This depicts the extent of post-harvest losses due to the perishability and deterioration of fruits before reaching the consumer, which occurs mostly after ripening of fruits [6].

The ripening of fleshy fruits is a developmental process that involves dramatic changes in color, texture, aroma, and nutrients, which provides essential carbohydrates, vitamins, minerals, phytonutrients, and fiber important for human and animal diets [7]. Fleshy fruit are categorized as climacteric and non-climacteric on the basis of their respiration and ethylene production during the onset of fruit ripening [8]. Climacteric fruits display a burst in respiration and biosynthesis of ethylene at the onset of ripening, in contrast with non-climacteric fruits which have no significant change in respiration and ethylene production during the transition from unripe to ripe [8, 9]. Several plant-associated microbes can also influence the regulatory steps of the ethylene pathway and ripening, by either directly producing ethylene or degrading ethylene produced by the host [10, 11].

Plant microbiomes consist of complex communities of potentially mutualistic, commensal, and pathogenic microbes colonizing the same niches in plants, including bacteria (such as *Bacillus* spp., *Staphylococcus* spp., *Pseudomonas* spp. and gram-negative bacilli, and yeasts and molds) that have diverse aspects of impacts on plant growth, health, and evolution [12, 13]. The microbes that constitute the fruit microbiome can colonize the fruit either vertically, as endophytes that live within plant tissues, or horizontally from the rhizosphere, or through mixed colonization [14].The fruit microbiota might be especially important during crop storage by preventing or favoring rots, or quality loss due to sprouting, water loss, or else spoilage.

Yet, pathogen-induced decay is certainly the most obvious reason for postharvest crop loss mainly caused by bacteria and molds [12].

Indeed, healthy fruits have many microbes on their surfaces but can usually inhibit their growth until after harvest [6]. Ripening weakens cell walls and decreases the amounts of anti-fungal chemicals in fruits, and physical damage during harvesting causes breaks in outer protective layers of fruits that spoilage organisms can exploit [13]. Molds are tolerant of acidic conditions and low water activity and are involved in spoilage of mango, banana, citrus fruits, papaya, apples, and other fruits. Molds such as *Penicillium*, *Botrytis*, *Aspergillus* and *Rhizopus* are frequently isolated from spoiled fruits. Yeasts and some bacteria, including *Cryptococcus*, *Candida*, *Enterobacteriaceae* and *Xanthomonas*, can also spoil some fruits [6]. *Botrytis* can cause a gray mold rot on many fruits and vegetables like banana, mango, citrus fruits, strawberries, tomato, onions, etc. This type of spoilage occurs in warm temperatures and high humidity, and is characterized by production of a gray mycelium on the fruit or vegetable [15].

According to a finding in Uganda [16], the microbial communities in banana fruits at different ripening stages are highly diverse but dominated by members of *Enterobacteriaceae*, *Pseudomonas* spp. followed by *Fusarium* spp; and the highest microbial load was observed at overripe stage of the fruits. Other authors [17] also reported that *Bacillus* spp., *Staphylococcus* spp., and *E. coli* were dominant in fruits. Likewise, the microbiome of banana and papaya fruits at different ripening stages were observed to be dominated by *Candida* spp., *Saccharomyces* spp. [18], *Aspergillus* spp. [17] and *Alternaria* spp. [19] Furthermore, fungal microbiome of orange samples was reported to constitute *Candida* spp., *Saccharomyces* spp. [20], *Aspergillus* spp. and *Penicillium*, spp. [21]. This result shows that the fruit at harvest stages were contaminated by key fungal postharvest pathogens which could result in spoilage of fruits.

The climatic and soil conditions of Ethiopia allow cultivation of a wide range of horticultural crops [22]. Fruit production and export plays a significant role in the local economy as a means of earning livelihoods for ∼5 million farmers, and for generating foreign exchange revenues [5]. More than 120 million hectares of land are under fruit crops in Ethiopia, with ∼55.11% of the area allocated to banana [4]. In 2018, ∼8.4 million quintals of fruits were produced in the country; while bananas, mangoes, papayas, and oranges took up 60.11%, 16.02%, 7.10%, and 4.94% of the fruit production, respectively [23]. However, more than 45% of the product is still lost due to poor post-harvest management [4]. This problem also sounds true to the Southwestern region of the country.

Southwestern Ethiopia, particularly the Bench-Sheko Zone is one of the evergreen, naturally rich parts of the Ethiopia providing surplus production of fruits and vegetables. Banana, orange, papaya, mango and avocado are among the major fruit crops produced throughout the year [22]. However, large quantity of fruits has been lost post-harvest, due to factors such as poor pre-and post-harvest handling, and poor transportation and storage conditions favoring massive loss of the fruits due to microbial-induced spoilage.

The study of microbiome profiles in fruits at different ripening stages is a hot issue of global researchers currently [13]. Previously, several studies have been conducted concerning microbiological quality and safety of fruits. Yet, previous studies have used only ready-to-eat fruit samples from market, or else, partially spoiled fruit samples to identify foodborne pathogens, or spoilage microbes, respectively; still with few parameters. The change in microbiome and physico-chemical profiles in fruits at different ripening stages was not/poorly studied. Therefore, the objective of the current study was to evaluate the change in microbiome profile of fruit samples of *Musa paradisiaca*, *Citrus sinensis* and *Carica papaya* at mature green, moderately ripe and overripe stages by determining microbial load, characterization and identification of any distinct bacterial and fungal colony encountered during microbial enumeration, using both conventional biochemical method and MALDTOF MS analysis; followed by

investigation of changes in physico-chemical properties of the fruit samples, by measuring the pH, moisture, titrable acidity, ascorbic acid, reducing sugar and total sugar contents of the fruits samples during ripening.

## Materials and methods

### Description of the study area

The study sites, Gura Ferda, Sheko, and Mizan-Aman Town and its surrounding districts, are located in the Bench-Sheko Zone which was formerly administered in the South Nations, Nationalities and Peoples Regional State (**S1 Fig**). Currently, the zone is reorganized under the newly established Southwestern National Regional State of Ethiopia (no online shapefile provided). The zone is situated at 6˚ 58' N and 35˚ 45'E, with altitudinal ranges between 900 and 1810m (masl). The mean minimum and maximum annual temperature of the area are 15.23˚C and 30.17˚C, respectively with the mean annual rainfall of the area being 1850.55 mm/year. The area is largely covered with tropical deciduous forest [24]. The three study sites were selected purposively; as the area is one of the well-known sites in Ethiopia for its high fruit production throughout the year [22].

### Study design

An experimental-based cross-sectional study design was employed. Samples were taken in three phases to obtain fresh fruit samples with different ripening stages.

### Sample size and sampling techniques

Totally, 108 fresh fruit samples, 36 bananas (*Musa paradisiaca*–Linnaeus, 1736), 36 oranges (*Citrus sinensis*–Swingle, 1943) and 36 papayas (*Carica papaya*–Linnaeus, 1753) were collected for this study. In the first phase of sample collection, 36 mature green fruit samples, 12 from each of the three fruit types, were collected and processed for both microbiological and physico-chemical analyses. This task was repeated for moderately ripe stage and overripe stage of the fruits. These stages were selected due to the assumption that fruits at these stages may exhibit significantly different Physico-chemical properties as well as changes in microbiome profile. The samples were collected separately and aseptically, put into sterile labeled polythene bags and transported using an icebox to Jimma University Research and Post-Graduate Laboratory for analysis. The samples were temporarily stored under refrigeration (4˚C) until processed.

### Sample preparation for microbiological analysis

An average of 1Kg of papaya, 600g of each banana, and orange fruit sample was weighed separately and aseptically chopped with its peel into small pieces using sterile blade and blended using an electronic blender (RL–Y66, Italy) until completely homogenized. Then, 25 g of each fruit juice was mixed in 225 mL of sterile peptone water and homogenized for 10 minutes using a shaker (Compact Shaker, D-72379 Hechingen, Germany). Then, serial dilution of each suspension was made up to $10^{-7}$. Finally, 0.1 ml aliquot of each dilution was spread-plated on a sterile pre-solidified agar plates and incubated at the appropriate temperature and time. Microbial enumerations were made in triplicates following standard methods [25].

**Microbial enumeration.** The viable colonies of total aerobic mesophilic bacteria (AMB), total aerobic spore-forming bacteria (ASFB), lactic acid bacteria (LAB), *Enterobacteriaceae*, Staphylococci [25], and yeasts and molds [26] and were enumerated in accordance with standard procedures. Briefly, 0.1 ml aliquot of each sample was spread plated in duplicates on Plate

Count Agar (PCA) (Oxoid), De Man Rogosa Sharpe (MRS) (Microgen), Violet Red Bile Glucose Agar (VRGBA) (Microgen), Mannitol Salt Agar (MSA) (Microgen), Potato Dextrose Agar (PDA) (Microgen) for the count of AMB, LAB, *Enterobacteriaceae*, Staphylococci, and molds/yeasts, respectively. For the enumeration of ASFB, a suspension of 10 mL was heated in a water bath at 80°C for 10 minutes to remove vegetative cells followed by inoculation of 0.1ml aliquot onto PCA and incubated at 30–32°C for 72hrs. Only plates containing a countable number of colonies (30–300 for bacteria and 10–150 for fungal colonies) were enumerated and the mean counts of the colonies were expressed in CFU ml$^{-1}$ which later converted to Log CFU ml$^{-1}$.

**Isolation and characterization of bacterial colonies.** The microbial communities (both epiphytes and endophytes) from all the fruit samples were counted, isolated, and characterized using standard methods [25]. After the enumeration of AMB from Plate Count Agar (PCA), 5–10 colonies of bacteria with distinct morphological differences such as color, size, and shape were randomly picked from countable plates and aseptically transferred into test tubes containing 5 ml Nutrient Broth (Microgen) and incubated at 32°C for 24 hours. Then, a loop-full of the broth growth was streak-plated on nutrient agar and incubated at 32°C for 24 hours. Sub-culturing was done until pure colonies were obtained, and the pure culture was preserved at 4°C as slants for further characterization. The pure cultures of the isolates were characterized for their colony morphology and biochemical reactions, and results were compared with the standard microbial properties in Bergey's manual of systematic bacteriology [27]. All the tests were performed in 3 replicates for each isolate to confirm the results. Finally, the identified isolates were preserved in 20% glycerol at -21°C.

**Morphological and biochemical tests.** The colony morphology (size, color, margin, and elevation), cellular morphology (Gram staining and motility test) of the isolates was characterized using standard methods [28].

**Gram staining.** The Gram staining was used basically to differentiate Gram-positive and Gram-negative bacteria, beside observing cellular shape and arrangement patterns. Gram-positive appear purple, and the Gram-negative bacteria appear pink. The staining was done using a standard method [28].

**Motility and H$_2$S production test.** Motility of the bacteria was determined by stabbing isolates in Sulfide-indole-motility (SIM) medium tubes with straight wire loop. Then, all the tubes were incubated at 37°C for 24 hours, with screw-cap kept loosened. Motile isolates shown diffused growth whereas non motile isolates were grown straight, and H$_2$S producing isolates shown dark growth [29].

**Triple sugar fermentation.** Triple Sugar Iron (TSI) agar, a differential medium was used to assess the ability of a microbe to ferment sucrose, glucose, and lactose in addition to produce gas and/or hydrogen sulfide. Importantly, TSI agar was used to identify enteric bacteria [25].

**Catalase test.** The presence of catalase activity in the isolates will be checked by flooding a drop of 3% H$_2$O$_2$ on a pure 24-hour colony over the slide, immediate effervescence of gas bubble was recorded as a positive result [25].

**Citrate test.** A 24-hour colony of an isolate was picked and lightly streaked on surface of Simmon's citrate agar slant and incubated for 24-48 hours. Citrate positive growth converts the medium from deep green to intense Prussian blue. The citrate negative isolates show trace/ no growth, and the medium remains deep green [25].

**Indole test.** A tube of Tryptone broth was inoculated with a small amount of an active colony and incubated at 37°C for 24 hours. To test for indole production, 5 drops of Kovác's reagent was directly added to the tube. A positive indole test was indicated by the formation of a "cherry-red ring" on top layer of the medium within seconds of adding the reagent [29].

**Oxidase test.** A small piece of Whatman No 1 filter paper was soaked in 1% (w/v) diethyl-amide benzaldhyde, and loop-full of 24-hour culture was picked and rubbed on the filter paper. The appearance of blue color within 10 and 30 seconds was considered as a positive test [25].

**Urease test.** A heavy inoculum from a 24-hour pure culture was streaked on entire surface of urea agar slant, and incubated with loosened caps at 35˚C. The slants were observed for a color change at 6 hours, 24 hours, and every day for up to 6 days. Urease production is indicated by a bright pink color on the slant that may extend into the butt, and any degree of pink is considered a positive reaction [25].

**Coagulase test.** The tube coagulase test was performed by mixing a pure 24-hour colony into 3mL plasma in a small test tube and incubated at 35˚C. The formation of clotting within 24 hours was considered as a positive result [25].

**Identification of dominant isolates using MALDI-TOF MS.** Representatives of suspected bacterial colonies formerly identified by the conventional biochemical method from all fruit samples were also analyzed using MALDI-TOF MS. The MALDI-TOF MS spectrometer (EXS3000 MALDI-TOF MS, China) was used to obtaining the mass spectra. The formic acid extraction procedure was achieved based on the manufacturer's guidelines for the identification of the bacterial isolates. Briefly, exact 300μl ultra-pure water was pipetted into 1.5mL centrifuge tube, 5-10mg of fresh 24h bacterial colony was picked using pipette tip and mixed fully. Then, 900μl absolute ethanol (100%) was added into the tube and well mixed by vortex mixer for 20s, centrifuged at 12000 rpm for 3min, the supernatant poured out, centrifuged again for 1min and the supernatant pipetted out carefully. After 5 minutes of drying the sample completely, 20μl lysate (70% formic acid) was added into the tube and tip mixed. The gram-positive bacteria were left for 5 minutes for enough lysate reaction. The, 20ul of lysate I (Acetonitrile, CAN) was added, mixed by Vortex mixer for 10s and centrifuged at 12000 rpm for 2 min. Finally, 1μl of the supernatant was pipetted onto target plate point, dried fully, after which 1μl of matrix solution was added to cover the sample, and died fully to test.

Similar to other authors [30], the spectra were produced by applying the Compass Satellite software, and the Microflex LT device was used for fast and accurate identification of bacterial colonies isolated from the fruit samples. A spectra score of 2.300–3.000 indicates that the identification of bacterial species has high reliability, 2.000–2.299 indicates the identification of possible species, 1.700–1.999 indicates the identification of possible genus, and 0.000–1.699 indicates that the identification result is unreliable. *E. coli* ATCC 8739 was used as the standard strain before the detection of each test sample. The analysis was repeated twice for each isolate and the result with better spectra score was recorded as a data.

**Isolation and characterization of mould and yeast colonies.** After enumeration on PDA plates, 5–7 yeast and mould colonies of different morphology, were aseptically transferred to separate 5 mL of PDB tubes, and PDA plates, respectively. A pure culture of each colony on each medium was obtained by repeated sub-culturing. The pure cultures of molds were aseptically transferred onto PDA slants, and incubated at room temperature again for 3 days, and stored at 4˚ C for further use. The pure cultures of yeasts were aseptically transferred onto PDB and incubated at 30˚C for 48 hours after which stored at 4˚C for further identification [26]. Then, the pure isolates of molds and yeasts were identified by microscopic observation of hyphal structure (molds), and sporangiospore and cellular structure by staining using Lacto-phenol cotton blue, using a taxonomic key as a morphological reference [31]. The *Candida* spp. were then screened using BiGGY Agar and germ tube test [26]. All the tests were performed in triplicate for each isolate to confirm the results. After identification, the isolates were preserved in 20% Glycerol at -20˚C.

## Determination of physico-chemical properties of fruit samples

**Sample preparation.** The samples collected directly from the local farms of the Bench-Sheko residents were transported to Jimma University Research and PG Lab. Each sample was separately and aseptically soaked into sterile water and washed. Then, the fruit samples were separately chopped into small pieces (with peel) and blended using an electronic blender (RL-Y66, Italy) so as to make composite juice of each fruit type. After vigorously shaken for 10 minutes using a shaker (Compact Shaker, D-72379, Germany), the juices were stored at 4°C for further physico-chemical tests [32].

**Peel color.** Banana peel color was considered for classification using the standard ripeness classification of bananas [33]. This scale categorizes bananas into seven stages of ripening: 1 = mature green; 2 = light green; 3 = half yellow–half green; 4 = three-quarters yellow with green; 5 = yellow with green tips; 6 = full yellow; and 7 = yellow with brown spots. For this study, three stages of ripeness were considered: stage 1 (mature green), stage 5 (yellow with green tips), and stage7 (yellow with brown spots) were considered for the collection of banana samples. In the case of orange, a specific color scale has been developed from 1 (deep orange) to 8 (dark-green), for a rind coloration of fruits, using an objective color charts reported by different authors [34]. In this study, the scores 7 (light green), 4 (yellow with trace of green) and 1 (deep orange) stages were selected [35]. For papaya, mature green (fully green), moderately ripe (yellow with trace of green), and overripe (deep orange) were considered [32].

**Determination of pH.** To determine the pH content of the fruit samples, exactly 10g of each fruit sample was placed in a 50mL beaker containing 12mL distilled water, and blended using an electronic homogenizer (RL–Y66, Italy) until smooth. Then, pH was measured using a digital pH meter (pH–013, China), pipetting 10ml of the homogenized sample into a beaker [36].

**Moisture content.** To determine the moisture content, clean oven-dried beakers of 50 ml capacity was weighed and recorded (W1). Then, 10gm of each sample was separately homogenized, weighed, and added into pre-weighed dried beaker (W2) and oven-dried at 105°C for 2 h. After 2 h of drying, the sample was relocated to a desiccator, and weight (W3) was recorded as in AOAC Official Method 934.06 [37]. Percentage of water content was determined as a ratio of differences between W2 and W3 to W2 and W1 multiplied by 100 as given below. Furthermore, total solid content was determined by subtracting the % moisture content from the total (100%).

$$\% \text{ of Moisture Content} = \frac{W2 - W3}{W2 - W1} \times 100$$

$$\text{Total Solid Content} = 100 - \text{Moisture content}$$

**Titrable acidity (TA).** The TA of fruits samples was determined by separately homogenizing 5 gm of each sample in 50 ml distilled water and filtered through Whatman No.1 filter paper. Then, about 9ml of the homogenate was pipetted into a beaker into which 3 drops of 1% phenolphthalein indicator were added titrated with 0.1 N NaOH solutions until pink color was observed. The TA was calculated as percent acid as described earlier in AOAC-Official-Method-942.15 [38] using the formula given below. Finally, the TA is calculated as % of mallic acid for banana and % of citric acid for orange and papaya as follows:

$$\% \text{ of Titrable acidity} = \frac{N \ NaOH \ X \ VNaOH \ X \ F \ acid}{Volume \ of \ Jucie \ used} \times 100$$

Where: $N_{NaOH}$ is the normality of NaOH used (g L$^{-1}$), $V_{NaOH}$ is the volume of NaOH solution consumed (L), $F_{acid}$ is a factor equivalent weight of the acid in the fruit juice sample equal to 0.067 for citric acid and 0.064 for mallic acid.

**Ascorbic acid content.** The ascorbic acid content in each fruit sample was determined by titration using 0.1 M iodine solution. For each sample, 25 mL of blended juice was poured into a 250 mL volumetric flask, and 10 drops of a pre-prepared 1% soluble starch solution was added. The mixed solution was titrated with 0.1 M of iodine solution until a blue color was observed to persist for 15 s. Before determination of ascorbic acid content of the fruit samples, the titration method was optimized using standard solutions of ascorbic acid, which was titrated with 0.1 M iodine solution. Each measurement was performed in triplicates, and quantified in mg per 100 g of fruit juice [39].

$$\text{Ascorbic Acid (mg/100g)} = \frac{\text{VI X FAA}}{\text{Mass of the juice sample used}} \times 100$$

Where: $V_I$ = the volume of iodine solution consumed (L), $F_{AA}$ = a conversion factor of the ascorbic acid consumed with iodine solution (molarity of iodine times the molecular mass of ascorbic acid).

**Determination of reducing sugars and total sugars.** The reducing and total sugar contents of all samples was determined using Lane and Eynon titration method, in accordance with AOAC Official Method 923.09 [40] and FSSAI [41]. Accordingly, to determine the Fehling factor, 4.75g of pure sucrose was weighed and transferred to 500 ml volumetric flask with 50 ml distilled water. Then, exactly 5 ml of 99% HCl was added and allowed standing for 24 hrs. The solution neutralized with 0.1N NaOH using phenolphthalein as end point indicator and made up to volume with distilled water. Then, the reaction was mixed well and 25 ml of the solution was transferred to a 100 ml volumetric flask and made up to volume with distilled water (1 ml = 2.5 mg of invert sugar). Then, the solution was transferred to a burette having an offset tip and titrated against Fehling's solution until the blue color disappears to a brick-red, and the titer was noted as $V_1$. Therefore,

$$\text{Fehling Factor (g of Invert Sugar)} = \frac{\text{V1} \times 2.5}{1000} = \text{V1} \times 0.0025$$

Where: W = Weight of the sample

$V_1$ = Volume of sucrose solution (titer) required for complete Fehling's reduction

Secondly, to determine the reducing sugar content, 10g of each sample was accurately weighed and transferred to 500 ml volumetric flask containing 100 ml distilled water and neutralized with 0.1N NaOH solution to phenolphthalein end point. Then 10 ml of neutral lead acetate solution was added, mixed well and let stand for 10 min. Then, potassium oxalate solution was added drop by drop until there is no further precipitation. The solution was mixed well and made up to volume with distilled water, and filtered through Whatman No. 1 filter paper. The filtrate was finally transferred to a 50 ml burette having an off-set tip. Then, to conduct the preliminary titration, 5 ml each of Fehling A and B solutions was pipette into 250 ml conical flask, mixed in 10 ml water and a few glass beads was added. Then, the sugar solution from the burette was added drop wise and the solution was heated to boiling, and 3 drops of methylene blue added as an indicator. The addition of the sugar solution drop wise was continued until the blue color disappears to a brick-red endpoint within boiling period of 3 minutes, and the titer value was noted down as $V_2$. For final titration, 5 ml each of Fehling A and B solutions was pipetted into a 250 ml conical flask and sample solution about 1.0 ml less than titer value in the preliminary titration ($V_2$) was added. The flask was heated to boiling, and titration

proceeds as above. The titer value was noted down as $V_3$. The titration was performed in duplicate and the average was recorded. Based on the factor for Fehling's solution, $V_3$ ml sample solution contains $0.0025\ V_1$ g reducing sugar (as invert sugar). Therefore,

$$\text{Reducing sugar in the sample (\%)} = \frac{0.0025 \times V1 \times V2}{V3 \times W} \times 100$$

Where: W = Weight of the sample
$V_1$ = Volume of sucrose solution (titer) required for complete Fehling's reduction
$V_2$ = Dilution volume for the sample (V1−1 ml)
$V_3$ = Volume of clarified sample solution (titer) required for Fehling's reaction

Thirdly, to determination of total reducing sugars, an aliquot of 50 ml of the clarified, sample filtrate was pipetted into a 100 ml volumetric flask, 5 ml of conc. HCl was added and allowed standing at room temperature for 24 hours. Then, neutralized with 0.1N NaOH solution using phenolphthalein as indicator, and made up to volume with distilled water and transferred to 50ml burette having an offset tip. Then, the titration was performed as in reducing sugars, and the titer noted as $V_4$. The total reducing sugars in $V_4$ could be calculated as $V_4$ ml = $0.0025 \times V_1$ g. However, as 50 ml of the sample solution is diluted twice (50 ml to 100 ml) after hydrolysis, dilution volume of the sample is $2V_2$. Therefore,

$$\text{Total Reducing sugar in the sample (\%)} = \frac{0.0025 \times V1 \times 2V2}{V4 \times W} \times 100$$

Finally, the amount of sucrose and total sugar in the sample was calculated as:

$$\text{Sucrose in the sample (\%)} = (\% \text{ Total reducing sugars } - \% \text{ Reducing sugars}) \times 0.95$$

$$\text{Total sugars in the sample (\%)} = (\% \text{ Reducing sugars } + \% \text{ Sucrose})$$

## Data analysis

The obtained data were statistically analyzed using the SPSS version 20 software package. The microbial enumeration was performed for each fruit type at three different ripening stages; and the colony counts were first calculated in CFU/ml and later converted to log CFU/ml. The physico-chemical tests were done in triplicate for each sample. Then, the normality and homogeneity of variance of the raw data was tested and confirmed $P > 0.05$, before the one-way ANOVA was run. Finally, the mean values (± standard deviation) of the replicated data were analyzed, and differences among the mean values obtained from the three fruit ripening stages were evaluated using one-way ANOVA: Duncan and Post Hoc multiple comparison methods; and the data with $P < 0.05$ were considered as significantly different.

## Ethical permission issues

The fruit samples required for this study were directly purchased under normal condition from farm sites of local farmers in districts of Gura Ferda, Sheko, and Mizan-Aman Town and its surrounding, Bench-Sheko Zone, Southwestern Ethiopia. A research approval and support letter from Jimma University, written to all concerned bodies was obtained and reported to the local administrations of Bench-Sheko Zone and respective districts of the study sites. All concerned bodies have approved the work site access. However, no written approval letters were required since the study involved only fruit samples from the farm sites, and did not related to medical, veterinary or socio-demographic aspects of the residents.

## Results

### Microbial enumeration in fruit samples at different ripening stages

The microbial load of the fruits at different stages of maturity: mature green, ripe and overripe showed a significant difference (P < 0.05). Accordingly, the maximum AMB was observed in overripe banana (7.96 ± 0.34 Log CFUg$^{-1}$) followed by over-ripe papaya (7.82 ± 0.32 Log CFUg$^{-1}$) while the minimum count was observed in mature green orange (6.74 ± 0.48 Log CFUg$^{-1}$). Relatively lower counts were recorded for *Enterobacteriaceae* (1.78 ± 1.57– 3.34 ± 0.28 Log CFUg$^{-1}$) and Staphylococci (2.06 ± 1.53 to 3.73 ± 0.29 Log CFUg$^{-1}$) in the three fruits. Overall, the microbial load increased with progress in the fruit's maturity stages from mature green through ripe to overripe with a significant difference (P < 0.05) between the stages. However, there is no significant difference (P > 0.05) in counts of *Enterobacteriaceae* in bananas and orange, and yeast and molds in orange fruit samples at different ripening stages (Table 1).

### Biochemical and MALD-TOF MS identification of bacterial isolates

In this study over-all, 243 bacterial isolates were identified using the conventional biochemical methods, and 36 representative isolates by MALD-TOF MS technique. Among nine categories of morphologically different bacterial colonies isolated in this study, seven of them were identified at genera level using the biochemical methods. These include *Bacillus* spp., *Pseudomonas* spp., *Staphylococcus* spp., *E. coli*, *Salmonella* spp., *Shigella* spp. *and Serratia Species*. Two of them remained unidentified by the conventional method, and later identified as *Alcaligenes faecalis* and *Morganella morganii*, using MALD-TOF MS technique.

In the present study, 36 bacterial isolates from top-five most dominant strains were randomly selected as representative, among which 97.2% of them were identified to species/genera level using MALDI-TOF MS. Then, various species of the isolates were recognized by matching their spectra with spectral profiles stored in the MALDI Biotyper database. Accordingly, 27 (75%) of the bacterial isolates were correctly identified at species level, by MALDI-TOF MS fingerprinting with a score value ≥2.00, 8 (22.2%) isolates were identified at genera level with a score value between 1.70–1.99, and only one isolate (2.8%) assumed to be *Alcaligenes* spp. has shown a score value <1.700. In this study, 6 species of bacteria were identified by the MALD-TOF MS, including *B. cereus*, *A. faecalis*, *P. putida*, *M. morganii*, *S. sciuri* and *S. epidermidis* (S1 Table). In addition, the spectra of representative isolates created by the MALDI Biotyper Compass Software that differentiate the strains at Genus/species level with distinct peak positions were illustrated below (S2 Fig).

### Distribution of dominant bacterial isolates in fruit samples

This study revealed that the bacterial microbiome of all fruit types at different ripening stages was majorly predominated by soil and environmental bacteria. Generally, a total of 243 bacterial isolates grouped into 9 genera were identified from 108 fruit samples collected for this study (S1 and S2 Files). The bacterial species identified using the morphological and biochemical tests, as well as the MALD-TOF MS analysis include *B. cereus* (33.7%), *A. faecalis* (17.3%), *P. putida* (15.2%), *M. morganii* (11.1%) and *S. sciuri* (6.6%), found in all fruit samples with different proportions. However, some bacterial species were exceptionally not detected in some fruits at certain maturity stage. For instance, the opportunistic pathogen *S. epidermidis*, and *E. coli* was not detected on all fruits at mature green stage. Similarly, *Salmonella* spp., *Shigella* spp. and *Serratia* spp. was not encountered in banana and papaya at moderately ripe stage and in orange at both mature green and moderately ripe stages. The *Serratia* spp. was not detected

**Table 1. Mean microbial counts (Mean Log CFU ml$^{-1}$ ± standard deviation) in fruit samples, Bench-Sheko Zone, 2021.**

| Microbial Groups | Banana (Musa paradisiaca) | | | | Orange (Citrus sinensis) | | | | Papaya (Carica papaya) | | | |
|---|---|---|---|---|---|---|---|---|---|---|---|---|
| | S1 | S2 | S3 | P value | S1 | S2 | S3 | P value | S1 | S2 | S3 | P value |
| TAMB | 6.76± 0.42$^c$ | 7.60 ± 0.42$^b$ | 7.96 ± 0.34$^a$ | 0.000 | 6.74 ± 0.48$^c$ | 6.81 ± 0.45$^b$ | 7.51 ± 0.43$^a$ | 0.000 | 6.79 ± 0.47$^c$ | 7.25 ± 0.33$^b$ | 7.82 ± 0.32$^a$ | 0.000 |
| TASFB | 4.77 ± 0.39$^c$ | 6.07 ± 0.38$^b$ | 6.87 ± 0.29$^a$ | 0.000 | 4.26± 0.51$^c$ | 5.41 ± 0.35$^b$ | 6.07 ± 0.11$^a$ | 0.000 | 4.63 ± 0.47$^c$ | 5.52 ± 0.42$^b$ | 6.24 ± 0.33$^a$ | 0.000 |
| LAB | 4.77 ± 0.71$^c$ | 5.61 ± 0.59$^b$ | 6.10 ± 0.39$^a$ | 0.000 | 4.95 ± 0.25$^b$ | 5.33 ± 0.53$^a$ | 5.68 ± 0.42$^a$ | 0.001 | 5.06 ± 0.27$^c$ | 5.76 ± 0.39$^b$ | 6.14 ± 0.15$^a$ | 0.000 |
| Entero | 2.32 ± 1.73$^a$ | 2.08 ± 1.54$^a$ | 2.93 ± 1.53$^a$ | 0.423 | 1.87 ± 1.65$^a$ | 1.78 ± 1.57$^a$ | 2.09 ± 1.55$^a$ | 0.884 | 2.75 ± 1.29$^{ab}$ | 1.86 ± 1.65$^b$ | 3.34 ± 0.28$^a$ | 0.019 |
| Staph | 2.08 ± 1.55$^b$ | 3.25 ± 0.38$^a$ | 3.37 ± 0.29$^a$ | 0.003 | 2.17 ± 1.62$^b$ | 2.21 ± 1.66$^b$ | 3.49 ± 0.36$^a$ | 0.036 | 2.06 ± 1.53$^b$ | 3.02 ± 1.02$^a$ | 3.73 ± 0.29$^a$ | 0.002 |
| Yeasts & moulds | 3.87 ± 0.43$^c$ | 4.38 ± 0.47$^b$ | 4.97 ± 0.24$^a$ | 0.000 | 3.74 ± 0.48$^a$ | 3.92 ± 0.39$^a$ | 4.03 ± 0.43$^a$ | 0.251 | 4.09 ± 0.23$^b$ | 4.21 ± 0.37$^b$ | 4.79 ± 0.41$^a$ | 0.000 |

Where, AMB = total aerobic mesophilic bacteria, ASFB = total aerobic spore-forming bacteria, LAB = lactic acid bacteria, Entero = Enterobacteriaceae, Staph = Staphylococcus spp., S1 = Mature green, S2 = Moderately ripe, S3 = Overripe stage. Different letters along with figures in a column indicate a significant difference (P < 0.05).

in all maturity stages of orange, and in banana and papaya at mature green. From this result, it could be concluded that the microbial load and diversity in all fruit samples was increasing with ripening; as the lowest microbial load and diversity was observed at mature green stage and the highest at overripe stage of the fruits (Table 2).

## Distribution of dominant fungal isolates in fruits at different ripening stages

In this study, a total of 375 fungal isolates (molds and yeasts) grouped into 9 genera were isolated from 108 fruit samples and identified using morphological and biochemical tests (**S1 and S2** Files). The results of this study shown that the fungal microbiome of the fruit samples at

**Table 2. Distribution of dominant bacterial species colonizing fruits at different ripening stages, Bench-Sheko Zone, 2021.**

| Fruit Sample | No. of Sample | B. cereus | A. faecalis | P. putida | M. morganii | S. sciuri | S. epidermidis | E. coli | Salmonella spp. | Shigella spp. | Serratia spp. | Total (%) |
|---|---|---|---|---|---|---|---|---|---|---|---|---|
| Banana S1 | 12 | 8 (9.8) | 2 (4.8) | 2 (5.4) | 3 (11.1) | 1 (6.3) | - | - | 1 (14.3) | 1 (16.7) | - | 18 (7.4) |
| Banana S2 | 12 | 11 (13.4) | 4 (9.5) | 3 (8.1) | 3 (11.1) | 2 (12.5) | 3 (25) | 1 (10) | - | - | 1 (25) | 27 (11.1) |
| Banana S3 | 12 | 15 (18.3) | 5 (11.9) | 5 (13.5) | 4 (14.8) | 2 (12.5) | 2 (16.7) | 2 (20) | 2 (28.6) | 1 (16.7) | 1 (25) | 39 (16.1) |
| Orange S1 | 12 | 4 (4.9) | 4 (9.5) | 4 (10.8) | 2 (7.4) | 1 (6.3) | - | 1 (10) | - | - | - | 16 (6.6) |
| Orange S2 | 12 | 6 (7.3) | 6 (14.3) | 5 (13.5) | 2 (7.4) | 1 (6.3) | 2 (16.7) | - | - | - | - | 22 (9.1) |
| Orange S3 | 12 | 9 (11) | 8 (19) | 8 (21.6) | 3 (11.1) | 2 (12.5) | 3 (25) | 2 (20) | 1 (14.3) | 1 (16.7) | - | 37 (15.2) |
| Papaya S1 | 12 | 7 (8.5) | 3 (7.1) | 3 (8.1) | 3 (11.1) | 2 (12.5) | - | 1 (10) | 1 (14.3) | 1 (16.7) | 1 (25) | 22 (9.1) |
| Papaya S2 | 12 | 10 (12.2) | 4 (9.5) | 3 (8.1) | 3 (11.1) | 3 (18.75) | 1 (8.3) | - | - | - | - | 24 (9.9) |
| Papaya S3 | 12 | 12 (14.6) | 6 (14.3) | 4 (10.8) | 4 (14.8) | 2 (12.5) | 2 (16.7) | 3 (30) | 2 (28.6) | 2 (33.3) | 1 (25) | 38 (15.6) |
| Total | 108 | 82 (33.7) | 42 (17.3) | 37 (15.2) | 27 (11.1) | 16 (6.6) | 12 (4.9) | 10 (4.1) | 7 (2.9) | 6 (2.4) | 4 (1.6) | 243 (100) |

Where, S1 = Mature green, S2 = Moderately ripe, S3 = Overripe stage, Number of isolates in columns = Frequency (%).

different ripening stages was predominated by *Candida* spp. (33.9%), *Saccharomyces* spp. (18.1%), *Aspergillus* spp. (16.3%), *Alternaria* spp. (8.5), *Penicillium* spp. (7.2%) and *Fusarium* spp. (6.7%); which were detected in all fruit samples with different proportions (Table 3). Still, the frequencies of isolation of other fungi differs in different maturity stages of the fruits. More specifically, *Alternaria* spp. (18.7%) were dominant in mature green papaya, while *Penicillium* spp. (22.2%) in overripe orange and *Fusarium* spp. (24%) dominated the overripe banana. Overall, the least number of *Mucor* spp. (3.2) and *Rhizopus* spp. (2.7) were observed in the tested fruit samples. Whereas, *Fusarium* spp. was not detected in mature green orange, as the case is for *Botrytis* spp. in mature ripe papaya, and moderately ripe and overripe banana. This result indicates that similar to the bacterial community, the fungal load and diversity in all fruit samples was increasing with ripening; as the lowest fungal load and diversity was observed at mature green stage, and the highest at overripe stage of the fruits (Table 3).

## Physico-chemical properties of fruit samples at different ripening stages

**Peel color.** In accordance with the standard rind color scale that categorizes the ripening level of banana, only three of the seven stages: stage 1 (mature green), stage 5 (yellow with green tips), and stage 7 (yellow with brown spots) was considered in this study. In the case of orange, the scores 7 (light green), 4 (yellow with trace of green) and 1 (deep orange) stages were selected [34]. For papaya fruit samples, mature green, moderately ripe (yellow with a trace of green), and overripe (deep orange) were considered. Peel color of the samples was matched to a standard reference keys and documented by photographing (**S3 Fig**).

**Physico-chemical composition of fruit samples.** The present study revealed that as ripening stage increased, the fruits showed significantly decreased pH value and increased TA and moisture content. Among the fruit samples, relatively orange was more acidic (pH, 3.53–4.03) than banana (pH, 4.47–5.73) and papaya (5.72–6.08). Moreover, the highest TA (1.28 mg/100g) was recorded in mature green oranges (Table 4). All the tests were conducted in triplicates and the mean value ± SD was analyzed (S2 Table).

On the other hand, the moisture content (%) increased as the ripening stage progressed from mature green (71.40 ± 0.95–87.27 ± 0.97) to overripe (75.77 ± 0.38–92.37 ± 0.95) in all

**Table 3. Distribution of dominant mold and yeast isolates in fruit samples at different ripening stages, Bench-Sheko Zone, 2021.**

| Sample Type | Sample size | *Candida* spp. | *Saccharomyces* spp. | *Aspergillus* spp. | *Alternaria* spp. | *Penicillium* spp. | *Fusarium* spp. | *Botrytis* spp. | *Mucor* spp. | *Rhizopus* spp. | Total (%) |
|---|---|---|---|---|---|---|---|---|---|---|---|
| Banana S1 | 12 | 13 (10.2) | 8 (11.8) | 6 (9.8) | 3 (9.4) | 2 (7.4) | 3 (12) | 2 (15.4) | 4 (33.3) | 2 (20) | 43 (11.5) |
| Banana S2 | 12 | 14 (11) | 8 (11.8) | 8 (13.1) | 2 (6.3) | 1 (3.7) | 4 (16) | - | - | - | 37 (9.9) |
| Banana S3 | 12 | 16 (12.6) | 9 (13.2) | 9 (14.8) | 5 (15.6) | 3 (11.1) | 6 (24) | - | 2 (16.7) | 2 (20) | 52 (13.9) |
| Orange S1 | 12 | 12 (9.5) | 5 (7.4) | 4 (6.6) | 2 (6.3) | 2 (7.4) | - | 2 (15.4) | - | 1 (10) | 28 (7.5) |
| Orange S2 | 12 | 13 (10.2) | 6 (8.8) | 5 (8.2) | 4 (12.5) | 4 (14.8) | 1 (4) | 2 (15.4) | - | - | 35 (9.3) |
| Orange S3 | 12 | 13 (10.2) | 7 (10.3) | 7 (11.5) | 3 (9.4) | 6 (22.2) | 3 (12) | 3 (23.1) | 1 (8.3) | 1 (10) | 44 (11.7) |
| Papaya S1 | 12 | 13 (10.2) | 7 (10.3) | 6 (9.8) | 6 (18.7) | 1 (3.7) | 1 (4) | 3 (23.1) | 1 (8.3) | - | 38 (10.1) |
| Papaya S2 | 12 | 16 (12.6) | 9 (13.2) | 8 (13.1) | 3 (9.4) | 3 (11.1) | 3 (12) | - | - | 2 (20) | 44 (11.7) |
| Papaya S3 | 12 | 17 (13.4) | 9 (13.2) | 8 (13.1) | 4 (12.5) | 5 (18.5) | 4 (16) | 1 (7.7) | 4 (33.3) | 2 (20) | 54 (14.4) |
| Total | 108 | 127 (33.9) | 68 (18.1) | 61 (16.3) | 32 (8.5) | 27 (7.2) | 25 (6.7) | 13 (3.5) | 12 (3.2) | 10 (2.7) | 375 (100) |

Where, S1 = Mature green, S2 = Moderately ripe, S3 = Overripe stage, Number of isolates in columns = Frequency (%).

**Table 4. Physico-chemical analysis (mean ± standard deviation) of the fruits at different ripening stages, Bench-Sheko Zone, 2021.**

| Component | Banana (Musa paradisiaca) | | | | Orange (Citrus sinensis) | | | | Papaya (Carica papaya) | | | |
|---|---|---|---|---|---|---|---|---|---|---|---|---|
| | S1 | S2 | S3 | P Value | S1 | S2 | S3 | P Value | S1 | S2 | S3 | P Value |
| pH | 5.73 ± 0.15[a] | 4.70 ± 0.10[b] | 4.47± 0.06[c] | 0.000 | 3.53 ± 0.06[b] | 3.63 ± 0.06[b] | 4.03 ±0.21[a] | 0.007 | 6.08 ±0.07[a] | 5.72 ± 0.07[b] | 5.67 ± 0.04[b] | 0.000 |
| TA (mg/100g) | 0.46 ± 0.04[a] | 0.91 ± 0.02[b] | 1.03 ± 0.08[c] | 0.000 | 1.28 ± 0.04[a] | 0.92 ± 0.02[b] | 0.75 ±0.03[c] | 0.000 | 0.03 ± 0.01[c] | 0.15 ± 0.01[a] | 0.10 ± 0.01[b] | 0.000 |
| Moisture (%) | 71.40 ± 0.95[a] | 74.33 ± 0.40[b] | 75.77 ± 0.38[c] | 0.000 | 86.20 ± 0.60[b] | 88.73 ± 0.35[a] | 85.53 ±0.83[b] | 0.002 | 87.27 ± 0.97[b] | 90.73 ± 1.60[a] | 92.37 ± 0.95[a] | 0.006 |
| SC (%) | 28.60 ± 0.95[a] | 25.67 ± 0.40[b] | 24.03 ± 0.72[c] | 0.001 | 13.80 ± 0.60[a] | 11.27 ± 0.35[b] | 14.47 ±0.83[a] | 0.002 | 12.73 ± 0.97[a] | 9.23 ± 1.60[b] | 7.63 ± 1.14[b] | 0.006 |
| AA (mg/100g) | 6.32 ± 0.08[b] | 8.10 ± 0.22[a] | 6.40 ± 0.05[b] | 0.000 | 39.77 ± 0.91[b] | 49.50 ± 0.70[a] | 40.77 ±1.39[b] | 0.000 | 61.63 ± 0.85[c] | 68.13 ± 0.31[b] | 69.87 ± 0.31[a] | 0.000 |
| RS (%) | 0.00 ± 0.00[c] | 12.13 ± 0.40[b] | 14.20 ± 0.40[a] | 0.000 | 4.07 ± 0.07[c] | 5.90 ± 0.26[b] | 8.87 ±0.31[a] | 0.000 | 4.63 ± 58[b] | 6.07 ± 0.03[a] | 6.30 ± 0.05[a] | 0.002 |
| TS (%) | 0.01 ± 0.01[c] | 15.20 ± 0.40[b] | 17.87 ± 0.31[a] | 0.000 | 6.80 ± 0.40[c] | 10.65 ± 0.79[b] | 12.73 ±0.61[a] | 0.000 | 5.22 ± 0.45[c] | 7.01 ± 0.20[b] | 7.88 ± 0.02[a] | 0.000 |

Where, AA = Ascorbic acid, TA = Titrable acidity, SC = Solid content, AA = Ascorbic Acid, RS = Reducing sugar, TS = Total Sugar, S1 = Mature green, S2 = Moderately ripe, S3 = Overripe stage. Different letters along figures indicate a significant difference (P < 0.05); the same letters indicate that there were no significant difference (P > 0.05) among the data.

fruits. The highest content of ascorbic acid (mg/100g) was observed in papaya (61.63–69.87) followed by orange (39.77–49.50) and banana (6.32 8–8.10). Also, significant increment in the amounts of total sugar (0.01 ± 0.01–17.87 ± 0.31) and reducing sugar (0.00 ± 0.00–14.20 ± 0.40) were recorded in banana and doubling in the amount of total sugar (6.80 ± 0.40–12.73±0.61) in orange with progress in the ripening stages (Table 4).

## Discussions

The microbial enumeration data in the present study showed that there was a continuous increment in the microbial load as ripening progresses in the fruit samples. The mean minimum and maximum counts (Log CFUg$^{-1}$) of AMB were noted at the mature green and overripe stages of all fruit samples. Accordingly, mean count (Log CFUg$^{-1}$) recorded from banana (6.76–7.96), and orange (6.74–7.51) were in consistence with previous reports [42]. The increasing microbial load in fruits in due course of maturation is probably due to the continuous increment of nutritional content of the fruits and time of exposure to vectors associated with fruits [17].

In the present study, 36 bacterial isolates from top-five most dominant strains were randomly selected as representative, among which 97.2% of them were identified to species/genera level using MALDI-TOF MS. Accordingly, the bacterial community of all fruit samples at different ripening stages was observed dominated by *B. cereus* (33.7%), *A. faecalis* (17.3%), *P. putida* (15.2%), *M. morganii* (11.1%), *S. sciuri* (6.6%) and *S. epidermidis* (4.9%). Majority of the identified isolates in this study were soil and environmental bacteria. In agreement with our observation in this study, it was also reported that *Bacillus* spp., *Staphylococcus* spp., and *E. coli* were dominant in fruits [17]. The highest number of *Bacillus* spp., and *Staphylococcus* spp. recorded in the study could be due to versatile metabolism, cross-contamination with soil carried by insects and birds vectors, and humans during farming [43]. The presence of enteric microorganisms in fruits could be due to the addition of human fecal matter and animal

manure in the traditional farm sites, and bird and insect vector interaction (use fruit plants as a nest) [17].

Moreover, the fungal microbiome of fruit samples at all ripening stages was dominated by *Candida* spp. (33.9%), followed by *Saccharomyces* spp. (18.1%) and *Aspergillus* spp. (16.3%). Likewise, fungal microbiome of orange samples was reported to constitute *Candida* spp. [17], *Saccharomyces* spp. [20], *Aspergillus* spp. and *Penicillium*, spp. [19]. Furthermore, different ripening stages of banana and papaya fruits were observed to be dominated by *Candida* spp., *Saccharomyces* spp., *Aspergillus* spp., and *Alternaria* spp. [43]. This result shows that the fruit at harvest stages were contaminated by key fungal postharvest pathogens which could result in spoilage of fruits [13]. Fruits can harbor relatively high numbers of microorganisms at harvest as normal microflora because of their potential contact with water and soil during growth [43].

In this study, the pH values the fruits were significantly decreased with progresses in maturity as observed in banana (5.73 to 4.47) and papaya (6.08 to 5.67) while TA values increased significantly as clearly observed in banana (0.46%, at mature green, to 1.03% at overripe stage). The finding is in line with an earlier report [44]. The same increment patterns were observed for papaya (from 0.03% to 0.15% at mature green and overripe stage, respectively), results supported by previous reports [32]. In contrary, significant decline (P < 0.05) in TA values with progress in maturity stages (1.28% to 0.75%, at mature green and overripe stage, respectively) were observed in orange samples. This was in line with a former findings [45]. As a result, the pH value of the orange samples has significantly increased (P < 0.05), with values ranging between 3.53 (mature green stage) to 4.03 (overripe stage). This finding is consistent with an earlier finding [46]. In citrus fruits, the cause for the decrease in acidity during the ripening might be due to the utilization of citric acid in the fruit respiratory process [46]. Fruits at green stage mainly contain starch, though the degradation of starch with progress in ripening may result in increase in sugars along with malic acid, citric acid and oxalic acid. This, in turn, results in an increase in TA content and decrease in pH values [47].

The mean moisture content in the analyzed banana fruits showed a significant increase (P < 0.05) with ripening stage (71.40%, at mature green, to 75.77% at overripe stage). This result is similar to the findings reported by Phillips et al., [48]. In papaya samples, too, it increased from 87.27% (at mature green) to 92.37% at overripe stage. This finding is in line with the result reported in Taiwan [49] and Malaysian papaya [50]. The high tendency of retaining and rise in moisture contents of these fruits is due to their ability to keep moisture content longer than other plants [50]. On the other hand, the moisture content of orange fruit samples showed a significant increase (P < 0.05) as ripening progressed from mature green (86.20%) to moderately ripe (88.73%) stage, but decreased at overripe stage (85.53%). This result is consistent with the findings of Gloria et al. [46] who conducted the analysis on Valencia oranges. The decrease in moisture content after mid stage of ripening is due to the onset of new fruit growth and development which suck back the moisture and nutrients from the old fruits of orange plant [45].

Ascorbic acid content in banana samples was significantly increased (P < 0.05) with progress in maturity from (6.32 mg/100g) mature green to moderately ripe stage (8.10 mg/100g) but decreased later at overripe stage (6.40 mg/100g). Similar results were also reported by other authors [36]. Likewise, the ascorbic acid content of orange increased with progress from mature green (39.77 mg/100g) to moderately ripe stage (49.50 mg/100g) though decreased significantly to 40.77 mg/100g at overripe stage. The same pattern in content of ascorbic was reported by Gloria et al., [46] in a study conducted on Valencia orange. The decrease in ascorbic acid content after mid-stage of ripening is due to enzymatic reactions and formation of dehydro-ascorbic acid from the oxidation of ascorbic acid, during over ripening [36]. In contrast the above two fruits, the Bench-Sheko papaya fruits presented continuous and significant

changes (P < 0.05) in ascorbic acid content during ripening stage from mature green (61.63 mg/100g) through moderate ripe (68.13 mg/100g) to overripe stage (69.87 mg/100g). This finding is consistent with an earlier report [51] indicating the ascorbic acid content would significantly increase with progress in maturity of papaya.

The Bench-Sheko banana samples showed a significant increase (P < 0.05) in both reducing and total sugar contents, as ripening progressed. The reducing sugar and total sugar contents were, respectively, 0.00% and 0.01% (mature green), 12.13% and 15.2% (moderately ripe) and 14.20% and 17.87% (overripe stage). Other studies also reported related dynamics in sugar contents with advance in ripening stages of banana [43, 48]. Similarly, mean reducing sugar content in orange fruit ranged from 4.07% (mature green), through 5.90% (moderately ripe) to 8.87% (overripe stage), in agreement with other similar report [46]. The total sugar contents in orange were also significantly higher both during moderately ripe (10.65%) and overripe (12.73%) maturity stages as compared to the mature green stage (6.80%), consistent with another finding [46]. Likewise, the mean reducing sugar contents of papaya was increased from 4.63% at mature green, to 6.07% at moderately ripe and 6.30% at overripe stage, results in agreement with report made from papaya analyzed in Indonesian [52]. Moreover, the total sugar contents of papaya samples increased from 5.22% (mature green), through 7.01% (moderately ripe) to 7.88% (overripe stage). This result is strongly supported by a previous findings [52]. The increment in soluble sugars results from starch degradation after seed maturation which is attended by increased activity of sucrose synthase during fruit ripening [53].

## Conclusions

With progress in the ripening stages of all fruits analyzed in this study, the microbial counts, diversity of microbiomes, physico-chemical parameters, including ascorbic acid, and total carbohydrate contents changed significantly. The microbial counts were significantly increased with ripening stage in all fruits, except counts of *Enterobacteriaceae* in banana and orange and the fungal counts in orange fruit samples. Most importantly, this study revealed the association of diverse microbiome including human and fruit pathogens such as *B. cereus*, *A. faecalis*, *M. morganii*, *S. epidermidis*, *Candida* spp., *Aspergillus* spp., *Alternaria* spp., *Penicillium* spp., and *Fusarium* spp. in Bench-Sheko fruit samples. Bench-Sheko zone is well known for its production and distribution of fruits to different areas of the country. Hence, considering an appropriate management system starting from farming to collection, distribution, and consumption of the fruits is critically important to ensure safety of the consumers and minimize product losses.

## Supporting information

**S1 File. Mean microbial counts from fruit samples at different ripening stages.**
(RTF)

**S2 File. Morphological and biochemical characterization of the microbial isolates.**
(RTF)

**S1 Table. MALDI-TOF MS identified bacterial species.**
(RTF)

**S2 Table. Physico-chemical properties of fruit samples at different ripening stages.**
(RTF)

**S1 Fig. Map of the study area.**
(RTF)

**S2 Fig. MALD-TOF mass spectrum of the bacterial isolates.**
(RTF)

**S3 Fig. Peel color and maturity level of the fruits samples at different ripening stages.**
(RTF)

## Acknowledgments

The authors would like to thank Jimma University, College of Natural Sciences for funding; Jimma University, Department of Biology, Ethiopian Ministry of Education, and Mizan–Teppi University for facilitating the study to **DK**.

## Author Contributions

**Conceptualization:** Dawit Raga Kifle, Ketema Bedanie Bacha, Reda Nemo Hora, Lata Lachisa Likasa.

**Data curation:** Dawit Raga Kifle, Ketema Bedanie Bacha, Reda Nemo Hora, Lata Lachisa Likasa.

**Formal analysis:** Dawit Raga Kifle, Ketema Bedanie Bacha, Reda Nemo Hora, Lata Lachisa Likasa.

**Funding acquisition:** Ketema Bedanie Bacha.

**Investigation:** Dawit Raga Kifle, Reda Nemo Hora, Lata Lachisa Likasa.

**Methodology:** Dawit Raga Kifle, Ketema Bedanie Bacha, Reda Nemo Hora, Lata Lachisa Likasa.

**Project administration:** Ketema Bedanie Bacha.

**Resources:** Dawit Raga Kifle, Ketema Bedanie Bacha.

**Software:** Dawit Raga Kifle, Reda Nemo Hora.

**Supervision:** Ketema Bedanie Bacha.

**Validation:** Dawit Raga Kifle, Ketema Bedanie Bacha, Reda Nemo Hora, Lata Lachisa Likasa.

**Visualization:** Ketema Bedanie Bacha, Reda Nemo Hora.

**Writing – original draft:** Dawit Raga Kifle.

**Writing – review & editing:** Dawit Raga Kifle, Ketema Bedanie Bacha, Reda Nemo Hora, Lata Lachisa Likasa.

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
