## [Editor Report · Decision Letter 0]

24 Jul 2023

PONE-D-23-16408Evaluation of Microbiome and Physico-chemical Profiles of Fresh Fruits at Different Ripening Stages: Implication to Quality and Safety ManagementPLOS ONE

Dear Dr. Kifle,

Thank you for submitting your manuscript to PLOS ONE. After careful consideration, we feel that it has merit but does not fully meet PLOS ONE’s publication criteria as it currently stands. Therefore, we invite you to submit a revised version of the manuscript that addresses the points raised during the review process.

We look forward to receiving your revised manuscript.

Kind regards,

Abhay K. Pandey

Academic Editor

PLOS ONE

Journal Requirements:

Additional Editor Comments:

The MS report an important findings and can be accepted after major revision.

---

## [Author Response · Author response to Decision Letter 0]

20 Sep 2023

All the comments and questions provided by both the Academic Editor and the Reviewer are carefully considered and incorporated accordingly. The modifications made in the manuscript are indicated as marked yellow in the file uploaded as "Revised Manuscript with Track Changes". The detailed information is uploaded labeled as "Response for reviewers".

---

## [Decision Letter · Decision Letter 1]

23 Oct 2023

PONE-D-23-16408R1Evaluation of microbiome and physico-chemical profiles of fresh fruits of Musa paradisiaca, Citrus sinensis and Carica papaya at different ripening stages: implication to quality and safety managementPLOS ONE

Dear Dr. Kifle,

Thank you for submitting your manuscript to PLOS ONE. After careful consideration, we feel that it has merit but does not fully meet PLOS ONE’s publication criteria as it currently stands. Therefore, we invite you to submit a revised version of the manuscript that addresses the points raised during the review process.

We look forward to receiving your revised manuscript.

Kind regards,

Abhay K. Pandey

Academic Editor

PLOS ONE

Additional Editor Comments:

Dear Authors The MS still needs major overhaul, please revise the MS in the light of referee comments.

Reviewers' comments:

Reviewer's Responses to Questions

**Comments to the Author**

1. If the authors have adequately addressed your comments raised in a previous round of review and you feel that this manuscript is now acceptable for publication, you may indicate that here to bypass the “Comments to the Author” section, enter your conflict of interest statement in the “Confidential to Editor” section, and submit your "Accept" recommendation.

Reviewer #1: All comments have been addressed

Reviewer #2: All comments have been addressed

2. Is the manuscript technically sound, and do the data support the conclusions?

Reviewer #1: Yes

Reviewer #2: Partly

3. Has the statistical analysis been performed appropriately and rigorously? 

Reviewer #1: Yes

Reviewer #2: Yes

4. Have the authors made all data underlying the findings in their manuscript fully available?

Reviewer #1: Yes

Reviewer #2: Yes

5. Is the manuscript presented in an intelligible fashion and written in standard English?

Reviewer #1: Yes

Reviewer #2: Yes

6. Review Comments to the Author

Reviewer #1: (No Response)

Reviewer #2: Ripening is a natural phenomenon in the fruits. Climacteric fruits should be harvested at fully mature stage whereas non climacteric fruits should be harvested after ripening at tree itself. I don't know what the objective behind this study was because all the fruits behave different. In case of citrus, how can one determine overripe stage. Please make it clear of ripening stage.

Second most important fact is that the microbial load over the fruit surface directly associated with the surrounding microclimate, harvest practices, and cultural operations which have been followed. There may be chances of contaminations at the time of harvesting but during handling operations we can minimize the contamination by several postharvest treatments. Why to go up to overripe stage. This is obvious that overripe fruits highly prone to microbial attack due to soft skin and less firmness along with poor defense system. The major infecting pathogens in each fruit already established by several researchers. Furthermore, author did not estimate defense related attributes as they play crucial role in delaying decay and infection process. So, please make it clear and establish a clear-cut hypothesis behind the research.

7. PLOS authors have the option to publish the peer review history of their article (what does this mean?). If published, this will include your full peer review and any attached files.

Reviewer #1: **Yes: **Dr. Nii Korley Kortei

Reviewer #2: **Yes: **Dr. Nirmal Kumar Meena, Agriculture University, Kota (India)

---

## [Author Response · Author response to Decision Letter 1]

25 Nov 2023

Comments and enquires from Reviewers, 

1. Climacteric fruits should be harvested at fully mature stage whereas non climacteric fruits should be harvested after ripening at tree itself. 

That is definitely right, and we did it. All the fruit samples at all ripening stages (including the non-climacteric orange fruit) used in this study were directly collected from farms of village residents, freshly cut from trees. That is why we collected samples by three phases of the needed ripening stages of the fruits (mature green, moderately ripe and overripe). All done aseptically. 

2. In case of citrus, how can one determine overripe stage? Please make it clear of ripening stage.

In order to obtain orange fruit samples at different ripening stages, we used rind coloration level. In case of citrus fruits, specific scores have been developed by using an objective color charts, from 1 (deep reddish orange) to 8 (dark-green) (le Roux 2006). In this study, orange fruits with color score 1 (deep reddish orange) was considered as overripe.

Reference: Smit le Roux, 2006. Preharvest manipulation of rind pigments of Citrus spp. Page 187.

3. Other most important fact is that the microbial load over the fruit surface directly associated with the surrounding microclimate, harvest practices, and cultural operations which have been followed. There may be chances of contaminations at the time of harvesting but during handling operations we can minimize the contamination by several postharvest treatments. Why to go up to overripe stage? 

As you have frankly said, the microbial load over the fruit surface directly associated with the surrounding microclimate, harvest practices, and cultural operations. This work is an original work us, since the microbiota and physico-chemical aspects of fresh cut fruits that can be affected by the surrounding microclimate, harvest practices, and cultural operations in the study area wasn’t assessed prior to this study. Additionaly, in this study, it is not only microbial load, but also alterations in microbial diversity and changes in physic-chemical properties of fruits at different ripening stages have been considered. The objective of this study was to evaluate the alterations in microbiome and physico-chemical properties of fresh cut fruits at different ripening levels (mature green, moderately ripe and overripe stages). At country level, we doubt that no study has been conducted at this level on microbiota of fruits.

Why we have gone up to overripe stage is to track on the alteration in microbiome profiles and physico-chemical properties of fruits at different ripening stages. This has also helped us to track the occurrences of human pathogens and the dominant spoilage microorganism causing fruit decay, and well identified. The application of green nanotechnology to alleviate loss of fruits and health problems occurring due to the presence of these microbes will be the next work of our project.

4. This is obvious that overripe fruits highly prone to microbial attack due to soft skin and less firmness along with poor defense system. The major infecting pathogens in each fruit were already established by several researchers. 

Previously, several studies have been conducted concerning only human and/or plant pathogens focusing only on fruit samples from markets, or partially spoiled fruit samples, still with limited parameters. The study of alterations in microbiome (microbial ecology) and physico-chemical profiles in fruits at different ripening stages was not conducted in depth, especially in Ethiopia.

5. Furthermore, author did not estimate defense related attributes as they play crucial role in delaying decay and infection process. So, please make it clear and establish a clear-cut hypothesis behind the research.

As a Ph.D. candidate, this manuscript is the first objective of my graduate dissertation. The project is of a wide scope, stretched up to preparation and application of food preservatives using green nanotechnology. Therefore, some in vitro and in vivo reactions of the identified microbes against the prepared nano-materials and the defense related attributes will be evaluated in the next series of study results under process.

---

## [Decision Letter · Decision Letter 2]

9 Jan 2024

Evaluation of microbiome and physico-chemical profiles of fresh fruits of Musa paradisiaca, Citrus sinensis and Carica papaya at different ripening stages: implication to quality and safety management

PONE-D-23-16408R2

Dear Dr. Kifle,

We’re pleased to inform you that your manuscript has been judged scientifically suitable for publication and will be formally accepted for publication once it meets all outstanding technical requirements.

Kind regards,

Abhay K. Pandey

Academic Editor

PLOS ONE

Additional Editor Comments (optional):

Dear Authors, at proof stage please address the following queries raised by reviewer.

Author has addressed all the queries however, there are minor corrections those need to be addressed. As I can see, somewhere it is mentioned 'log' and Log. likewise, level of significance (P< 0.05 or P> 0.05). References need to be recorrected and in similar format.

Reviewers' comments:

Reviewer's Responses to Questions

**Comments to the Author**

1. If the authors have adequately addressed your comments raised in a previous round of review and you feel that this manuscript is now acceptable for publication, you may indicate that here to bypass the “Comments to the Author” section, enter your conflict of interest statement in the “Confidential to Editor” section, and submit your "Accept" recommendation.

Reviewer #2: All comments have been addressed

2. Is the manuscript technically sound, and do the data support the conclusions?

Reviewer #2: Yes

3. Has the statistical analysis been performed appropriately and rigorously? 

Reviewer #2: Yes

4. Have the authors made all data underlying the findings in their manuscript fully available?

Reviewer #2: Yes

5. Is the manuscript presented in an intelligible fashion and written in standard English?

Reviewer #2: Yes

6. Review Comments to the Author

Reviewer #2: Author has addressed all the queries however, there are minor corrections those need to be addressed. As I can see, somewhere it is mentioned 'log' and Log. likewise, level of significance (P< 0.05 or P> 0.05). References need to be recorrected and in similar format.

7. PLOS authors have the option to publish the peer review history of their article (what does this mean?). If published, this will include your full peer review and any attached files.

Reviewer #2: **Yes: **Dr. Nirmal Kumar Meena

---

## [Editor Report · Acceptance letter]

22 Jan 2024

PONE-D-23-16408R2 

PLOS ONE

Dear Dr. Kifle, 

I'm pleased to inform you that your manuscript has been deemed suitable for publication in PLOS ONE. Congratulations! Your manuscript is now being handed over to our production team.

Kind regards, 

on behalf of

Dr. Abhay K. Pandey 

Academic Editor

PLOS ONE